# FAPbBr_3_ Perovskite Nanocrystals Embedded in Poly(L–lactic acid) Nanofibrous Membranes for Enhanced Air and Water Stability

**DOI:** 10.3390/membranes13030279

**Published:** 2023-02-26

**Authors:** Madeeha Tabassum, Qasim Zia, Jiashen Li, Muhammad Tauseef Khawar, Sameen Aslam, Lei Su

**Affiliations:** 1School of Engineering and Materials Science, Queen Mary, University of London, London E1 4NS, UK; 2Department of Materials, The University of Manchester, Oxford Rd., Manchester M13 9PL, UK; 3Department of Clothing, National Textile University Faisalabad, Faisalabad, Punjab 37610, Pakistan; 4Garments Technology Department, Punjab Tianjin University of Technology, Lahore 53720, Pakistan

**Keywords:** metal halide perovskite, polymers, perovskite nanocrystals, air stability, nanofibres

## Abstract

Formamidinium lead bromide (FAPbBr_3_) nanocrystals have emerged as a powerful platform for optoelectronic applications due to their pure green photoluminescence (PL). However, their low colloidal stability under storage and operation reduces the potential use of FAPbBr_3_ perovskite nanocrystals (PeNCs) in various applications. In this study, we prepared the poly(L–lactic acid) (PLLA) nanofibrous membrane embedded with FAPbBr_3_ perovskite nanocrystals by electrospinning the perovskite and PLLA precursor solution. This is a simple and low-cost technique for the direct confinement of nano-sized functional materials in the continuous polymer nanofibres. PLLA as a polymer matrix provided a high surface framework to fully encapsulate the perovskite NCs. In addition, we found that FAPbBr_3_ PeNCs crystallize spontaneously inside the PLLA nanofibre. The resultant PLLA-FAPbBr_3_ nanofibrous membranes were stable and remained in the water for about 45 days without any evident decomposition. The results of this research support the idea of new possibilities for the production of air-stable FAPbBr_3_ PeNCs by forming a composite with PLLA polymer. The authors believe this study is a new milestone in the development of highly stable metal halide perovskite-based nanofibres, which allow for potential use in lasers, waveguides, and flexible energy harvesters.

## 1. Introduction

Organic–inorganic hybrid perovskites have been a major area of interest within the field of efficient light-emitting sources [1]. Most recently, formamidinium lead bromide (FAPbBr_3_) NCs have received considerable attention due to their ultrapure green photoluminescence (PL) (530–535 nm) and higher thermal stability in comparison to more popular methylammonium lead bromide (MAPbBr_3_) [2]. These NCs are capped with long-chain ligands that enhance their stability in nonpolar or moderately polar organic solvents. However, perovskite nanocrystals (PeNCs) have poor stability against aqueous media, polar solvents, and long-term exposure to a moist environment [3]. Thus far, PeNCs stability was mainly achieved by alloyed perovskite nanostructures [4], core/shell-like structures [5,6], and encapsulation into polymer matrices [7,8]. Among them, the polymer matrix is the most widely used approach to synthesizing highly stable PeNCs. For instance, Cai et al. reported a new technique to synthesize polymer-assisted Cs-based perovskite nanocrystalline film. The polymer, poly(2-ethyl-2-oxazoline) (PEOXA), which coordinates with the metal cations in the perovskite, improves the phase stability of PeNCs and reduces the fabrication temperature. In addition, pure red perovskite light emitting diodes (LEDs) with maximum luminance of 338 cd/m^2^ and external quantum effeciency (EQE) of 6.55% were produced [9]. In another major study, Hou et al. reported the use of bolck copolymers that worked as nanoreactors during perovskite crystallization and passivated the NCs surface defects, hence improving its photostability in strong polar solvents [10].

Polymer nanofibres (PNFs) are one-dimensional long filaments with diameters in the range of tens to hundreds of nanometers. They offer unique properties of high porosity, bending elasticity, and high encapsulation capacity. Among several techniques to produce PNFs, electrospinning is the most cost-effective and efficient way to process a wide range of polymers [11]. Furthermore, the processing conditions of electrospinning offer advantages to encapsulate functional nanoparticles in the long fibres without post-treatments [12]. To date, numerous studies have attempted to produce perovskite–polymer hybrid NFs by employing an electrospinning approach. Wang and co-workers used the emulsion electrospinning technique to prepare ultrastable perovskite–polymer nanofibres [13]. In another extensive study by Tsai and colleagues, the one-step core/shell electrospinning method was used to prepare perovskite-based NFs. The uniform perovskite–polymer nanofibres showed the hydrophobic properties of shell polymer and were endowed with high resistance to water for 48 h [14]. Electrospinning is a simple technique to produce highly uniform nanofibres from a variety of polymers including poly(methyl methacrylate (PMMA), poly(vinylidene fluoride) (PVDF), poly(vinylpyrrolidone) (PVP), polyacrylonitrile (PAN), and polystyrene (PS). The excellent mechanical properties of some polymers, such as flexibility and easy preparation of PVDF–perovskite-based nanofibers, show significant advantages over rigid polymers (for example, PMMA, PS, etc.) [15,16]. Additionally, environmental issues related to different polymers in electrospinning have not been completely understood. Taken together, the current research studies highlight the importance of the use of biodegradable biopolymers, as well as green solvents.

Polylactic acid (PLA) is the most commonly used bioplastic. Lactic acid, the precursor of PLA, can be derived from various raw materials including corn, starch, and sugarcane, which is then polymerized to PLA. Among different kinds of PLA, poly (L–lactic acid) (PLLA) has been an object of many studies due to its biocompatibility, biodegradability, and eco-friendly nature [17]. It has proven to be a good alternative to petro-chemical based polymers for numerous purposes. In contrast to other biodegradable and bio-based polymers, PLLA show better processing parameters and can be processed by injection molding, extrusion techniques, ultrasonication, and fibre spinning. There are relatively few PLLA-based studies in the area of perovskites to make them stable at ambient conditions with low-cost processing methods. Yanyan et al. reported the first in situ synthesis of highly stable biopolymer phosphors with CH_3_NH_3_PbBr_3_ as NCs and PLLA as the matrix. The fabricated perovskite–polymer composite films showed uniform morphology with outstanding emission characteristics of PeNCs and high transmittance of PLLA matrices [18]. In another study, an ultra-flexible and transparent conductive substrate was fabricated from PLA with silver nanowires to advance green electronics [19]. The research to date, however, has not fully explored the potential of PLLA in the emerging field of perovskites.

The present research explores, for the first time, the use of PLLA nanofibres to encapsulate FAPbBr_3_ NCs. We mentioned in situ room temperature fabrication of FAPbBr_3_ NCs–PLLA nanofibres with FAPbBr_3_ NCs as an emitter and PLLA as the matrix. FAPbBr_3_ NCs–PLLA nanofibres can be easily produced by the electrospinning technique, which is low-cost, versatile, and allows for the control of NFs composition. The as-produced composite film shows a homogeneous morphology, combining the excellent properties of both materials. The outstanding properties of PLLA help to make highly stable and efficient electrospun nanofibrous films with excellent stability in ambient environments, retaining >50% of PL emission intensity. Overall, this study established a straightforward and low-cost method to prepare the composite film of FAPbBr_3_ NCs–PLLA for strong stability in air and water.

## 2. Experimental

### 2.1. Materials

Poly (L–lactic acid) (PLLA) (MW = 1.43 × 10^6^) was supplied by PURAC biochem, Holland. Chemicals including lead bromide (PbBr_2,_ ≥98%), formamidinium bromide (FABr, ≥98%), oleic acid (technical grade, 90%), oleylamine (technical grade, 70%), dimethylformamide (DMF, 99.8%), and dichloromethane (DCM, 99.99%) were purchased from Sigma-Aldrich.

### 2.2. Fabrication of Nanocrystals Embedded PLLA Electrospun NFs

The FAPbBr_3_ perovskite NCs incorporated in PLLA NFs were prepared by an electrospinning technique, as previously mentioned by Qasim et al., with some modifications [17]. PLLA (1.8%) was dissolved in 5 mL of dichloromethane in a closed glass vial by stirring and heating at 50 °C until the solution became clear and PLLA was completely dissolved. In parallel, PbBr_2_ (0.0376 g) and FABr (0.0124 g) were dissolved in 0.5 mL of DMF. Then, 200 µL oleic acid and 15 µL oleylamine were mixed and poured into the perovskite solution. Next, this perovskite precursor solution was added dropwise into a vigorously stirred PLLA solution at room temperature (25 °C) for 24 h. Then, the mixture was filled into a plastic syringe using a metal needle (19G) and was loaded into a syringe pump. The distance between the syringe tip and the collector was fixed at 30 cm and a high voltage of about 23 kV was applied to charge the precursor solution. The electrospinning solution was released towards the moving roller (200 rpm) to collect NFs. The collected NFs were allowed to dry overnight.

### 2.3. Synthesis of FAPbBr_3_ Perovskite NCs

We followed the ligand-assisted reprecipitation (LARP) method for the synthesis of FAPbBr_3_ perovskite NCs according to the literature, with reasonable changes [20]. PbBr_2_ (0.1 mmol) and FABr (0.1 mmol) were dissolved in 0.5 mL of DMF to prepare the precursor solution. Then, 250 µL of oleic acid and 15 µL of oleylamine were mixed separately in a vial and added to the solution. After that, 130 µL of this freshly prepared precursor solution was dropped into the 10 mL of chloroform under constant stirring and we observed the immediate formation of a green solution. Purification of PeNCs was completed by introducing a small amount of acetonitrile followed by centrifugation at 1200 rpm for 10 min. The collected PeNCs precipitate was further dispersed in 4 mL hexane for characterization.

## 3. Characterization

The morphology of FAPbBr_3_ NCs and FAPbBr_3_ NCs–PLLA nanofibres was examined by using TEM (JEOL-JEM, accelerating voltage 200 kV) and SEM (FEI Inspect F), respectively. The chemical composition of samples was determined by EDS (attached to FEI Inspect F) at a voltage of 10 kV. The confocal microscope was used to record the fluorescence images with a 488 nm laser as the excitation light source. X-ray diffraction analysis of as-prepared NFs and FAPbBr_3_ NCs was performed by PANalytical diffractometer with an angular range of 5 < 2θ < 70°.

The UV–Vis absorption and PL spectra were recorded using Perkin Elmer Lambda 35 UV–Vis spectrophotometer and Perkin Elmer LS55 spectrofluorometer, respectively. Fourier transform infrared (FTIR) spectra were recorded using a Burker model FTIR with a range of 4000–400 cm^−1^. The composition of as-prepared samples was confirmed by X-ray photoelectron spectroscopy (Thermo Scientific Nexsa XPS system) with an excitation source of AL Kα X-rays.

## 4. Results and Discussion

PLLA was used as a matrix for FAPbBbr_3_ NCs due to its outstanding resistance against water and ambient conditions. As schematically shown in Figure 1, we followed the electrospinning technique for the in situ ambient fabrication of FAPbBbr_3_ NCs–PLLA nanofibres. A precursor solution of PbBr_2_, FABr, and PLLA in DMF/DCM was used for electrospinning.

A liquid is electrified to generate a jet that was directed toward the grounded collector, and the solvent evaporated simultaneously during the electrospinning. The high electric potential between two electrodes able to overcome the surface tension inside the electrospinning solution allows the facile synthesis of continuous fibres. When the concentration of the electrospinning solution reached the critical saturation level, crystallization of the FAPbBbr_3_ NCs took place within the PLLA polymer.

### 4.1. Morphology of FAPbBbr_3_ NCs–PLLA Nanofibres

Figure 2a,b presents the scanning electron microscopy (SEM) images of PLLA NFs, which show the individual and highly uniform nanofibre structure. The morphological characterization of the composite film of FAPbBr_3_ NCs–PLLA nanofibres reveals the smooth nanofibre structure with the FAPbBr_3_ NCs embedded homogeneously in the PLLA polymer matrix, as shown in Figure 2c.

The average diameter of as-synthesized nanofibres is 1.45 µm, and not influenced by the incorporation of FAPbBr_3_ NCs. Many recent studies have shown that well-distributed NCs inside the polymer matrices do not affect the diameter of the nanofibres [1,21].

Respective high-resolution SEM images (Figure 2d) of FAPbBr_3_ NCs–PLLA nanofibres show that the fabricated PeNCs are well dispersed inside the PLLA matrix, and have a different shape, from cubic to spherical, depending on the amount of precursor solution concentration. Respective confocal fluorescence images exhibit strong PL emission (Figure 2e,f).

Scanning electron microscopy coupled with energy dispersive spectroscopy (EDS) was used for the analysis of FAPbBr_3_ NCs–PLLA nanofibres to observe the elemental composition. The results, as shown in Appendix A, indicate that both Pb and Br are present in the composite films.

Therefore, the above-mentioned discussions demonstrate that the FAPbBr_3_ NCs are homogeneously distributed in the PLLA matrix network through the in situ ambient fabrication technique.

FAPbBr_3_ NCs are synthesized with FABr/PbBr_2_ using the LARP technique and purified according to the previously mentioned report by Chen et al., with rational changes [20]. In this method, oleic acid (OA) and oleylamine (OLA) are used as surface ligands, where OA plays a key role to prevent the PeNCs from segregating and improving the colloidal stability of PeNCs. We investigated the optical properties of pristine FAPbBr_3_ NCs via UV absorption and photoluminescence (PL) measurements. In Figure 3a, the PL spectrum displays a pure-green emission at 530 nm. The PL emission spectra, with FWHM ≈ 22 nm are from pure FAPbBr_3_ NCs dispersion in hexane. From the normalized UV–Vis spectra, it demonstrated a strong light absorption edge at 512 nm.

Figure 3b shows the TEM images of the as-obtained FAPbBr_3_ NCs. The PeNCs crystallize into a cubic shape, monodispersed, with an average size of 10–12 nm. The PeNCs follow the typical perovskite structure ABX_3_, as shown in Figure 4b.

### 4.2. The Crystallinity of FAPbBbr_3_ NCs–PLLA Nanofibrous Membranes

The crystallinity of FAPbBr_3_ NCs was further analysed by X-ray diffraction (XRD) patterns.

The XRD patterns of PLLA nanofibres, FAPbBr_3_ NCs, and FAPbBr_3_ NCs–PLLA nanofibres are displayed in Figure 4a. Diffraction peaks located at 2θ = 14.8°, 16.8°, and 19.07° refer to PLLA nanofibres. The diffractogram of FAPbBr_3_ NCs (Appendix A) consists of major peaks at 2θ = 14.80°, 21.1°, 29.96°, 33.50°, 37.93, 42.81°, and 45.52° corresponding to the crystal planes of (100), (110), (200), (210), (211), (220), and (300), respectively. These diffraction peaks correspond to the cubic phase of PeNCs with the Pm–3m space group as reported in the literature [22,23]. In XRD patterns of FAPbBr_3_ NCs–PLLA nanofibres, the peaks of perovskite appear at 29.96°, 33.50°, 42.81°, and 45.52° which correspond to the (200), (210), (220), and (300) planes of FAPbBr_3_ NCs. Nevertheless, the diffraction peak intensity is very low due to the low concentration of FAPbBr_3_ NCs in the PLLA matrix, as well as the coverage effect of the PLLA matrix [18]. These findings show that high-quality FAPbBr_3_ NCs have been successfully fabricated inside the PLLA matrix.

### 4.3. Surface Chemistry of the FAPbBr_3_ NCs–PLLA Nanofibres

Fourier transform infrared spectroscopy (FTIR) has been widely used to study the chemical changes and conduct the quantitative analysis of polymers as well as serving to investigate the functional groups of the attached molecules. In Figure 5, FTIR spectra of PLLA nanofibres, FAPbBr_3_ NCs, and FAPbBr_3_ NCs–PLLA nanofibres are demonstrated. The spectrum shows bands at 2944 and 3000 cm^−1^ for PLLA, 2850 and 2924 cm^−1^ for PeNCs, and 2850 cm^−1^ for FAPbBr_3_ NCs–PLLA nanofibres. These bands belong to the C–H stretch from CH_3_. The presence of this stretching confirms the encapsulation of FAPbBr_3_ NCs into the polymer matrix. Compared to PeNCs bands at 3270 cm^−1^ for N–H stretching, there is negligible vibration for FAPbBr_3_ NCs–PLLA. This is due to the lower amount of PeNCs and coverage effects of the PLLA polymer matrix as confirmed by the X-ray diffraction patterns of these samples. In addition, the bands at 1745 cm^−1^ for PLLA and FAPbBr_3_ NCs–PLLA correspond to the C=O extension of the ester. For FAPbBr_3_ NCs, the C=O stretch can be ascribed to the presence of oleic acid at the surface of perovskite NCs.

X-ray photoelectron spectroscopy (XPS) is extensively employed to measure the binding energy and composition of constituents. As shown by the XPS spectra, it further confirms the presence of Pb and Br in the PLLA matrix, suggesting the formation of FAPbBr_3_ NCs inside the PLLA matrix. As can be seen in Figure 6a, elements of O (≈531 eV), C (≈ 284eV), Pb (≈143 and 138), and Br (≈69) were observed on the surface of FAPbBr_3_ NCs–PLLA nanofibres. However, there is no obvious peak of elements of N (≈401 eV) present on the composite film due to the lower concentration of PeNCs as confirmed by the XRD analysis. Regarding the Pb 4f and Br 3d spectra, the intrinsic peaks show the presence of encapsulated NCs inside the PLLA matrix (Figure 6b,c). In addition, the measured binding energies of different elements (Pb 4f, Br 3d) were very similar to those of FAPbBr_3_ NCs fabricated by other techniques.

### 4.4. Air and Water Stability of Composite Membranes

PeNCs can be embedded in a variety of polymers to fabricate nanocomposites that show an array of advantageous properties. The polymer matrix provides stability and flexibility while the PeNCs maintain their size, morphology, and composition-dependent characteristics. Polymers offer unique attributes, such as the ability to interact with the perovskites, low diffusion rates of moisture and oxygen, passivation of NCs surface defects, and decreased agglomeration of PeNCs in the solid state. The use of polymers provides strong chemical interaction with PeNCs, hence reducing the chances of moisture-induced degradation. In addition, the morphology and shape of polymer-encapsulated NCs do not change and PL intensity decreases considerably slower [24].

This section of the study was concerned with the stability of FAPbBr_3_ NCs–PLLA nanofibrous films in water and air. The PL peak position and intensity were changed slightly after 10 days of storage in air (Figure 7a), which showed that composite films are extremely stable in air. After 45 days of storage in ambient conditions, the PL peak intensity was still about 50% of its original value. Moreover, after 10 days of direct contact with water (Figure 7b), the PL intensity still retained 70% of the original value. In addition, there is a negligible shift in the PL peak of the FAPbBr_3_ NCs–PLLA nanofibrous films after storage in air and water. The change in PL intensity value as a function of storage time in air and water was summarized in Figure 7c. In Figure 7d,e, the photographs of PLLA nanofibres, FAPbBr_3_ NCs–PLLA nanofibrous films, and their water immersion is displayed. A comparison of the most recent research studies available on the FAPbBr_3_ PeNCs embedded into the polymers using the electrospinning method for water and air stability is summarized in Table 1. The encapsulation of PeNCs inside the PLLA nanofibres substantially enhances the stability against air and even under full immersion in water.

## 5. Conclusions

In summary, the first stable FAPbBr_3_ NCs–PLLA electrospun composite films were prepared in ambient or room temperature conditions and tested for air and water stability. We established that this technique is simple, low cost, and robust enough to fabricate FAPbBr_3_ NCs that are homogenously distributed into the PLLA polymer nanofibres. These PeNCs are effectively protected and exhibited similar stabilities against air and water stress as those mentioned in the previous studies. Moreover, more than 70% of the PL intensity is still maintained after 45 days of storage. The findings reported here shed new light on the use of biopolymers for optoelectronic applications including lasers, waveguides, and flexible energy harvesters. In addition, the flexible nature of nanofibres films may open the door for the production of flexible and bendable optoelectronics. However, PeNCs-based composite films with tunable emissions in the visible spectrum remain to be explored. Hence, this research will stimulate further study in the area of high-performance luminescence applications.

## Figures and Tables

**Figure 1 membranes-13-00279-f001:**
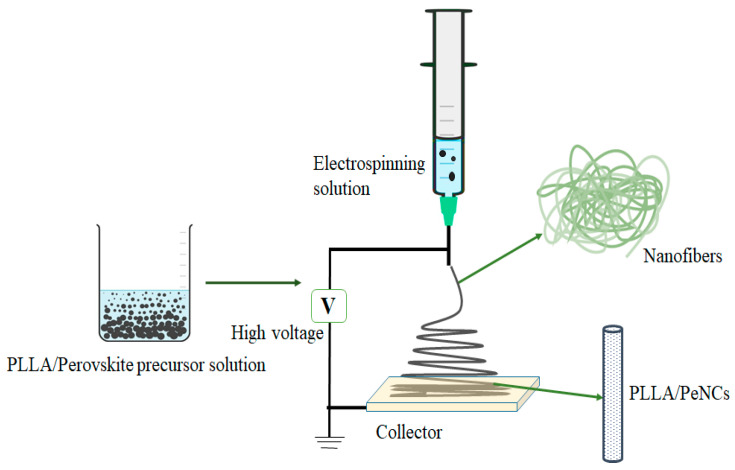
Schematic of electrospinning set-up for the fabrication of perovskite NCs –PLLA nanofibres.

**Figure 2 membranes-13-00279-f002:**
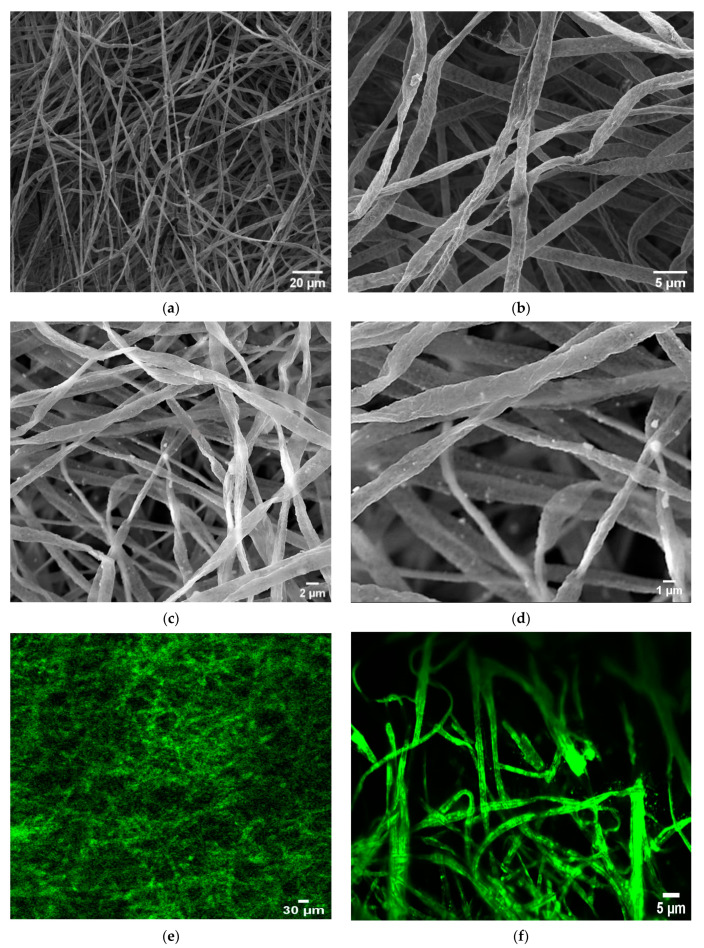
SEM images of the electrospun perovskite-polymer membranes. (**a**,**b**) Morphology of pristine PLLA nanofibres with different magnifications. (**c**,**d**) SEM image FAPbBr_3_ NCs@PLLA nanofibres (**e**,**f**) Confocal fluorescence microphotographs of perovskite–polymer membranes with a 488 nm laser as the light source.

**Figure 3 membranes-13-00279-f003:**
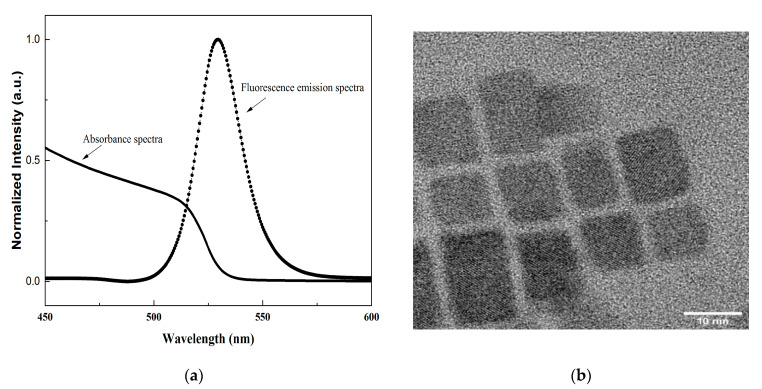
(**a**) UV–Vis absorption and PL spectra of FAPbBr_3_ NCs. (**b**) TEM images of FAPbBr_3_ NCs.

**Figure 4 membranes-13-00279-f004:**
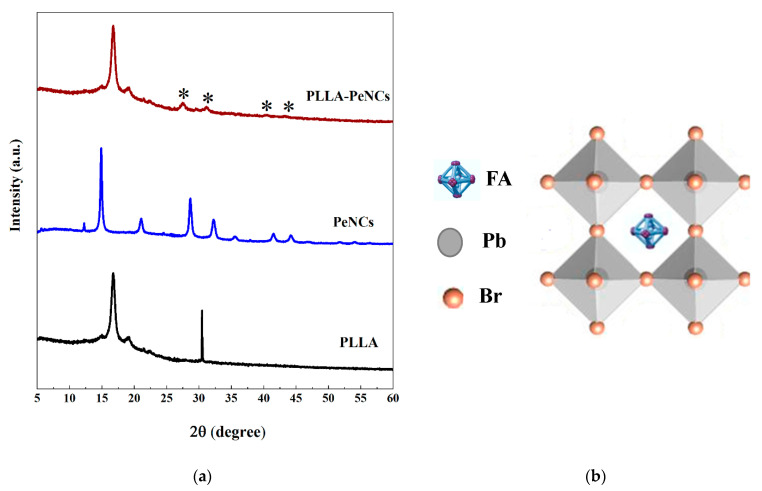
(**a**) X-ray diffraction pattern of pure FAPbBr_3_ NCs, PLLA, and PeNCs–PLLA nanofibres; * indicates the peaks belonging to FAPbBr_3_. (**b**) The cubic crystal structure of FAPbBr_3_ NCs at room temperature.

**Figure 5 membranes-13-00279-f005:**
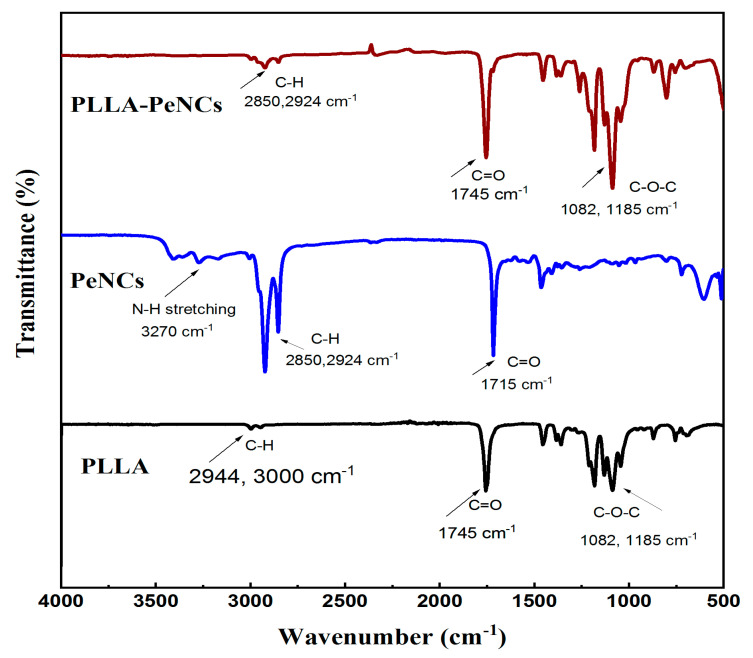
FTIR spectra of PLLA nanofibers, pure FAPbBr_3_ NCs, and perovskite nanocrystals encapsulated by PLLA nanofibres.

**Figure 6 membranes-13-00279-f006:**
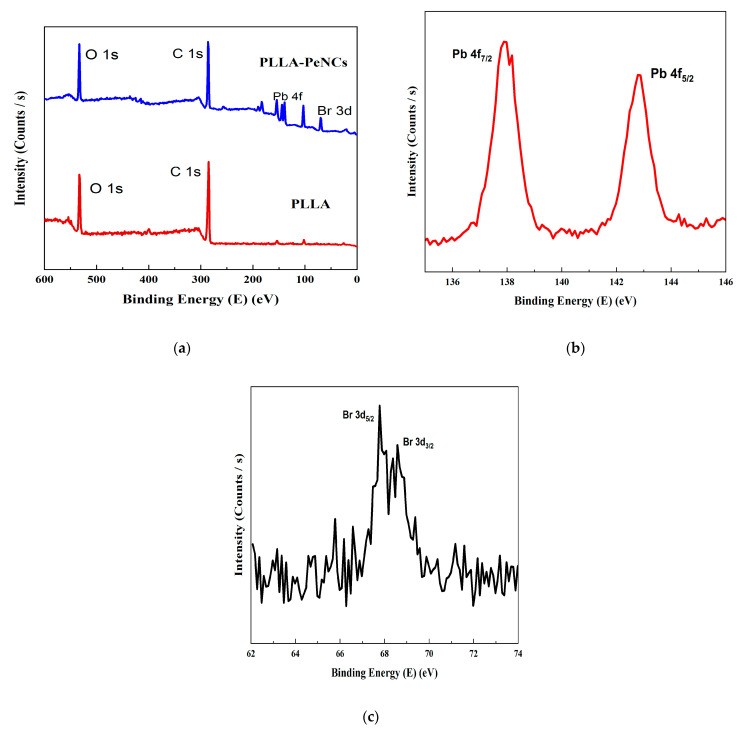
XPS analysis of (**a**) PLLA and PeNCs- PLLA nanofibres. (**b**,**c**) Pb 4f and Br 3d XPS spectra of composite nanofibres.

**Figure 7 membranes-13-00279-f007:**
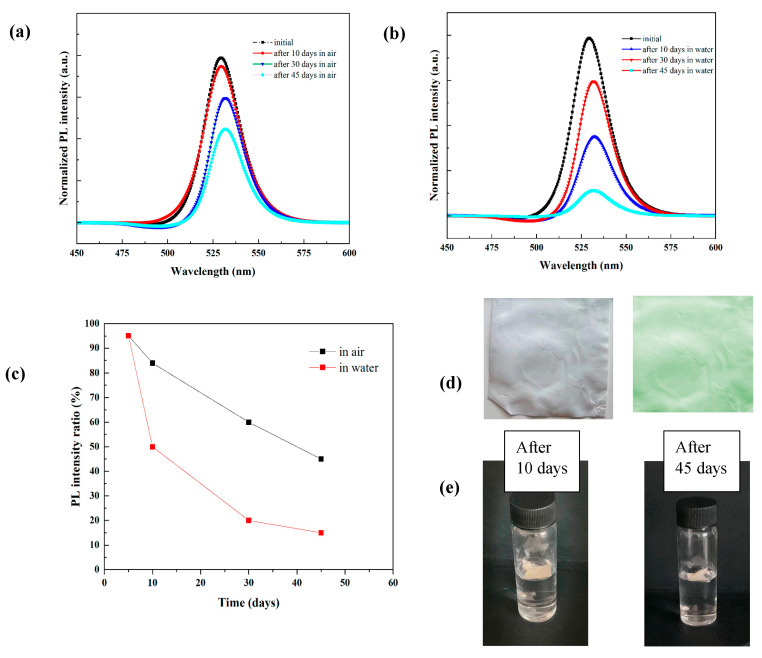
PL intensity variation spectra of FAPbBr_3_ NCs–PLLA nanofibres with storage time (**a**) in air and (**b**) in water. (**c**) Summary of PL intensity ratio for the composite films as a function of storage time in air and water. (**d**) Photographs of PLLA nanofibres and FAPbBr_3_ NCs–PLLA nanofibres. (**e**) Photographs of FAPbBr_3_ NCs–PLLA nanofibres after immersion in water after 10 and 45 days.

**Table 1 membranes-13-00279-t001:** Comparison of the FAPbBr_3_ perovskite NCs incorporated in poly (L–lactic acid) nanofibers with some other materials.

Materials	Polymer Used for Encapsulation	Fabrication Technique	Structure	PL intensity and Stability Parameters	Ref.
MAPbBr_3_ NCs	PM NFs MA, PVDF	Electrospinning	Core-shell NFs	10 days in the air ≈ 93%	[5]
CsPbBr_3_ NCs	PVP, PAN	Electrospinning	NFs	96 hr exposure in the air ≈ 73%	[6]
CsPbX_3_ (X = Cl, Br, and I) QDs	PVP	Electrospinning	NFs	4 days in water for green emitters	[7]
CsPbI_3_ QDs	PVDF	Electrospinning	NFs	3 days in water ≈ 80%	[4]
CsPbX_3_ (X = Cl, Br, and I) NCs	PAN	Electrospinning	Core-shell NFs	48 hr in water ≈ 50%	[8]
CsPbBr_3_ NCs	PVDF-PS	Electrospinning	NFs	70 days in water ≈ 90%	[9]
CsPbX_3_ (X = Cl, Br, and I) QDs	PS, PMMA, poly(styrene-butadiene-styrene) (SBS)	Electrospinning	NFs	3 months in water ≈ 80%	[10]
FAPbX_3_ (X = Cl, Br, and I) QDs	PS, PMMA	Microfluidic electrospinning	NFs	Consistent PL after 5 days immersion in water	[11]
CsCu_2_I_3_, Cs_3_Cu_2_I_5_	PS	Electrospinning	NFs	Consistent PL after 20 days immersion in water	[12]
FAPbBr_3_ NCs	PLLA	Electrospinning	NFs	45 days in the air ≈ 50%, 10 days in water ≈ 70%	

## Data Availability

The data presented in this study are available on request from the corresponding author.

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
