# Peer review of "FAPbBr3 Perovskite Nanocrystals Embedded in Poly(L–lactic acid) Nanofibrous Membranes for Enhanced Air and Water Stability"

_membranes, 2023, doi:10.3390/membranes13030279_

Round 1

Reviewer 2 Report

The reviewed paper presents the characterization and air and water stability tests for novel FAPbBr3 NCs-PLLA electrospun composite films. This paper is interesting and well written. The results are clearly described.

Author Response

Thank you for the feedback. 

Reviewer 3 Report

The manuscript describes the use of PLLA nanofibres to encapsulate FAPbBr3 NCs. The authors use the electrospinning technique which is low-cost, versatile and allows the control of the resulting materials. The composite films prepared present strong stability in air and in water.

Please, check the spelling of the text. 

For instance, Among instead of amon should be written.

Also, the flexible nature...instead of The flexible nature...

Author Response

Thank you for the feedback. 

We have corrected the grammatical errors throughout the manuscript.